# Comparing the physiques of elite Polish female and male swimmers training for short and long distances with their non-training peers – Is swimming a health-promoting sport?

Aleksandra Stachowicz[1], Anna Kęska[2]*, Katarzyna Milde[2], Małgorzata Stachowicz[3]

**1** Doctoral School, Józef Piłsudski University of Physical Education, Warsaw, Poland, **2** Department of Human Biology, Józef Piłsudski University of Physical Education, Warsaw, Poland, **3** Department of Water and Winter Sports, Józef Piłsudski University of Physical Education, Warsaw, Poland

* anna.keska@awf.edu.pl

## Abstract

In athletes, anthropometric measures are widely used to prescribe desirable body weight, to optimize competitive performance, and to evaluate the effectiveness of various training regimens. However, it also seems interesting to find out which values of anthropometric indices showing a significant relationship with health risk characterise the top athletes, especially in popular sports. The aim of the study was to characterise the physique of female and male swimmers compared to their non-training peers and to determine to what extent it is influenced by training distance. Somatic measurements were taken in 30 female and 30 male top Polish swimmers. The subjects were divided into four groups, i.e., SDF (n = 24) and SDM (n = 24) groups comprising females and males training for short-distance, and LDF (n = 6) and LDM (n = 6) groups comprising those training for long-distance. The swimmers were compared with their non-training peers, 373 females and 155 males aged 20–30 years. BMI, WHR, BF, BAI and Slenderness index were calculated to describe the athletes' physiques. Swimmers had significantly lower BMI (19.3 ± 1.4 in SDF and 22.1 ± 0.6 in LDF compared to 23.1 ± 3.7 in non-swimmers, $p < 0.001$) and WHR (0.7 ± 0.0 in SDF and 0.8 ± 0.0 in LDF, compared to 0.9 ± 0.1 in non-training peers, $p < 0.001$) and BF, but only in SDF group (22.7 ± 1.6, compared to 24.2 ± 5.9 in non-swimmers, $p < 0.01$). In contrast, athletes had significantly higher Slenderness index values (44.8 ± 1.1 in SDF, and 42.9 ± 0.5 in LDF, vs. 41.9 ± 3.6 in non-training peers, $p < 0.001$). It was also found that the distance trained differentiates especially the physique of female swimmers, and that physique has a greater impact on athletic performance in short-distance swimmers. Monitoring the physique of top swimmers provides an insight into the type of adaptation to the training process, which in turn enables the identification of factors that determine sports success. Such observations also document the health benefits of swimming, allowing this form of physical activity to be promoted to the public.

**Data availability statement:** All relevant data are within the manuscript.

**Funding:** The author(s) received no specific funding for this work. The research was conducted as part of a project entitled 'Single-Topic Research Task No. 3', carried out at the Józef Piłsudski University of Physical Education in Warsaw.

**Competing interests:** The authors have declared that no competing interests exist.

## Introduction

Swimming is a low-impact, full-body workout that promotes cardiovascular health, improves lung capacity, strengthens musculoskeletal structures, and reduces the risk of joint injuries commonly seen in high-impact sports [1]. The health benefits of swimming are largely due to changes in body composition and overall physique [2]. Regular swimming training usually induces an increase in muscle mass, especially in the upper body muscles, which generate most of the propulsive force and determine swimming speeds. Additionally, swimmers tend to develop a leaner physique with a low percentage of body fat, contributing to improved buoyancy and reduced drag in the water [3]. However, in the literature studies showing that swimmers have a higher body fat content than other athletes and non-athletes can also be found [4].

An important aspect of swimming training is its effect on body fat distribution. Unlike in many land-based sports, swimmers often retain a slightly higher level of subcutaneous fat, particularly around the torso, which may serve as insulation in the aquatic environment and aid in buoyancy [5]. As swimming is not only a popular sport, but also a type of physical activity that is important in the prevention of obesity and other chronic non-communicable diseases, it is of interest to determine how values of anthropometric indices showing a significant association with the risk of adverse health outcomes related to obesity characterise those who practice swimming compared to non-exercisers.

It is well known that the amount and distribution of body fat depends primarily on sex hormones. Males tend to have lower levels of body fat and a more central fat distribution compared to females [6]. Gender differences in fat content result in different its utilisation during exercise, as well as a different risk of developing diseases as a consequence of excess fat mass, particularly visceral fat [7]. Previous research has shown that gender influences changes in body composition in response to swimming training [8]. It therefore seems important to investigate to what extent swimming training modifies the amount and distribution of body fat in female and male swimmers.

Another factor that influences changes in physique in response to swimming training is the distance trained. Swimming can be performed at different distances. In pool swimming competitions the distances range from 50m to 1500m [9]. Research to date suggests that sprinters and long-distance swimmers often exhibit distinct morphological profiles that contribute to their specific competitive advantages. Sprinters focus on explosive power, anaerobic capacity, and high-intensity resistance training to maximize speed and strength [10]. In contrast, long-distance swimmers emphasize endurance, aerobic efficiency, and sustained energy output, often incorporating high-volume training with a focus on pacing and technique conservation [11]. These differences in training regimens influence the development of specific somatic characteristics that improve hydrodynamics and overall performance in short- and long-distance swimming. Understanding this relationship can provide valuable information on optimising training, predicting performance and long-term health implications for elite swimmers.

Previous studies on the body built of swimmers have mainly focused on its impact on the biomechanics of movement in water and athletic performance [12,13]. There

are far fewer studies analysing the changes caused by swimming training on anthropometric indicators, the values of which depend on lifestyle and indicate the risk of chronic non-communicable diseases [8,14]. In addition, there are few studies comparing the values of these indicators in swimmers and non-athletes [15]. This study aims to answer the question to what extent do the values of common anthropometric health-related indices in swimmers differ from their values in their non-training peers. An attempt was also made to examine the relationship between somatic characteristics and distance specialization among elite female and male swimmers.

## Materials and methods

### Participants

Ethical approval of the study was obtained from the Ethics Committee of the Józef Piłsudski University of Physical Education in Warsaw (No. SKE-01-15/2024). Recruitment for the study was conducted from 1 September to 30 October 2024. The inclusion criteria were high athletic level and participation in training sessions during the study, while exclusion criteria were injury preventing training or lack of consent to participate in the study. Participation in the study was voluntary, which each person confirmed by signing a written consent.

The study involved 30 female and 30 male swimmers aged 19–26 years (seniors), competing in pool swimming events. These were Polish athletes at the top of the national ranking lists at the time of the survey. The swimmers' performance has been expressed as FINA (Federation Internationale de Natation) point score, calculated according the formula '1000 x (world record time/athlete time)³ (s/s)³' from data available at https://www.swimrankings.net/?language=pl [16]. The training length of the study participants averaged 12 years. Over the study period, all athletes were completing 8–10 in-water training sessions a week (90–120 minutes), swimming a distance of 6,000–14,000 m per day. They also underwent land training 2–4 times a week (60–90 minutes), during which they performed strength, mobility and stabilisation exercises.

The subjects were divided into four groups based on their gender and distance specialisation. The SDF (n = 24) and SDM (n = 24) groups included female and male swimmers training for short distances, i.e., 50m and 100m, while the LDF (n = 6) and LDM (n = 6) groups included those training for long distances, i.e., 800m and 1500m. The general sport characteristics of the studied groups are shown in Table 1.

The swimmers' physiques were compared with those of their non-training peers. The control group consisted of a total of 528 subjects (373 females and 155 males) aged 20–30 years (22.2 ± 3.0 years) studied by Kopiczko et al. [17].

### Anthropometric measurements

Somatic measurements were carried out according to the protocol of the International Society for the Advancement of Kinanthropometry [18]. Body height, body weight, waist and hip circumferences were checked. Height was measured to the nearest 0.1 cm and weight to the nearest 0.1 kg. Measurements were taken using a stadiometer and a medical scale. Waist and hip circumferences were measured to the nearest 0.1 cm using a non-stretch measuring tape. Waist

**Table 1. Study groups of female (n = 30) and male swimmers (n = 30), mean ± SD.**

|  | Training length, years | FINA points |
|---|---|---|
| SDF (n = 24) | 11.5 ± 2.3 | 768.5 ± 50.9 |
| LDF (n = 6) | 11.8 ± 2.5 | 808.2 ± 50.9 |
| SDM (n = 24) | 13.3 ± 2.0 | 833.5 ± 56.4 |
| LDM (n = 6) | 12.5 ± 3.6 | 848.7 ± 45.3 |

Abbreviations: SDF – short-distance female swimmers, LDF – long-distance female swimmers, SDM – short-distance male swimmers, LDM – long-distance male swimmers.

circumference was measured at the mid-point between the lowest rib and the top of the iliac crest. Hip circumference was measured at the level of the greatest protrusion of the gluteal muscles.

The participants underwent identical assessment session in the morning hours. Before the testing, they were well rested, after fasting overnight, and properly hydrated. They were asked to avoid intense physical activity for at least 12 hours before the measurements. All tests were conducted in a well-ventilated test room with controlled temperature and humidity by a trained person with many years of experience.

### Anthropometric indexes

Based on the above measurements, five indexes were calculated to describe the athletes' physiques. The first was the body mass index (BMI) calculated according to the formula 'BMI = body mass/height$^2$ (kg/m$^2$)' and interpreted according to the World Health Organization classification. According to it, a BMI < 18.5 indicates underweight, BMI values 18.5–24.9 indicate normal weight, BMI values 25.0–29.9 indicates overweight and a BMI > 30.0 indicates obesity [19]. The waist-to-hip (WHR) index was calculated according to the equation 'waist circumference/hip circumference (cm/cm)'. Its values were also interpreted using World Health Organisation guidelines, according to which in men android (visceral-abdominal) fatness was assumed for WHR values ≥ 1.0, and gynoid (gluteal-femoral) for WHR values < 1.0. In women, android fatness is found when WHR ≥ 0.85, and gynoid fatness when WHR < 0.85 [20].

Two indexes assessing body fat percentage were also used, i.e., the Body Fat Index (BF) and the Body Adiposity Index (BAI). BF was calculated according to the formula 'BF = 1.2 × BMI + 0.23 × calendar age - 10.8 × gender - 5.4 (where men = 1; women = 0)' [21], and BAI as 'BAI = hip circumference/body height$^{1.5}$ – 18 (cm/m$^{1.5}$)' [22]. Interpretation of body fat content in athletes and non-athletes was made using the standards proposed by Gallagher et al. [23]. According to them in men, body fat content below 8% is considered low, values between 8% and 21% are considered normal, values between 21% and 26% are classified as overweight and values above 26% as obese. For women, a body fat percentage of 21% − 33% is considered normal, values below 21% are classified as underweight, values between 33% and 39% indicate overweight, while values above 39% are classified as obesity.

The formula 'body height/ $\sqrt[3]{\text{body weight}}$ (cm/kg$^{1/3}$)' was used to calculate the Slenderness index [24]. The classification of body types on the basis of this index is as follows: in men, index values ≤ 41.54 indicate a bulky build, values 41.55–44.87 indicate a medium build and values ≥ 44.88 indicate a slender build. In women, the cut-off values for the Slenderness index are: ≤ 41.47 blunt body build, 41.48–44.96 medium body build and ≥ 44.97 slender body build.

### Statistical analysis

The normality of the distribution of the variables was tested using the Shapiro-Wilk test. The significance of differences between the groups separated on the basis of gender and distance trained was assessed using a two-factor analysis of variance ANOVA. The existence of a relationship between somatic parameters/indicators and swimmers' sporting level was ascertained from the Spearman correlation coefficient. A significance level of $p < 0.05$ was assumed for all tests. Statistical calculations were performed using STATISTICA v.13.

### Results

It was observed that swimmers were taller than non-swimmers, both short- ($p < 0.001$) and long-distance ones ($p < 0.05$). Short-distance swimmers had lower body mass ($p < 0.001$), while long-distance swimmers had higher body mass ($p < 0.05$) than non-athletes. Both groups showed smaller waist (short-distance: $p < 0.001$; long-distance: $p < 0.05$) and larger hips ($p < 0.001$) compared to controls (Fig 1). It was also determined that swimmers had lower BMI ($p < 0.001$), BF (short-distance: $p < 0.001$; long-distance: $p < 0.01$), and WHR ($p < 0.001$), but higher Slenderness Index ($p < 0.001$) than non-swimmers. No differences were found between the compared groups (athletes/non-athletes) in the BAI (Fig 1).

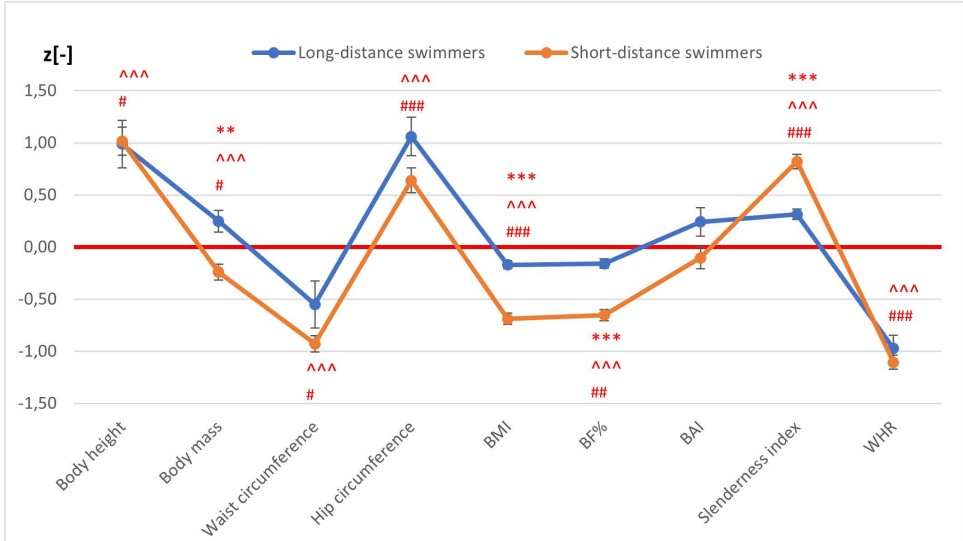

**Fig 1. Mean (± SE) values of anthropometric parameters and indexes of swimmers normalized to non-training groups.** * $p<0.05$, ** $p<0.01$, *** $p<0.001$ significant differences between the mean of the normalised values in short-distance and long-distance swimmers. ^ $p<0.05$, ^^ $p<0.01$, ^^^ $p<0.001$ significant differences between the mean of the normalised values in the short-distance swimmers and the control group. # $p<0.05$, ## $p<0.01$, ### $p<0.001$ significant differences between the mean of the normalised values in the long-distance swimmers and the control group. red line – mean values of the variables studied for non-trained females and males.

Table 2 shows that the groups of athletes did not differ in height. Among women, long-distance swimmers had greater body weight ($p<0.01$) and waist and hip circumference ($p<0.001$) than short-distance ones. No such differences were observed among male swimmers.

Two-factor ANOVA (gender × distance) revealed significant effects on all body composition indices (Table 3). For BMI, males ($p<0.001$) and long-distance swimmers ($p<0.001$) had higher values, with SDF group showing the lowest BMI (gender $F_{1,56}=42.80$; $p<0.001$; $\eta^2=0.43$; distance $F_{1,56}=22.23$; $p<0.001$; $\eta^2=0.27$; interaction $F_{1,56}=13.84$; $p<0.001$; $\eta^2=0.20$).

BF was higher in females ($p<0.001$) and long-distance swimmers ($p<0.001$), with the largest differences observed among female swimmers (gender $F_{1,56}=350.15$; $p<0.001$; $\eta^2=0.86$; distance $F_{1,56}=21.06$; $p<0.001$; $\eta^2=0.27$; interaction $F_{1,56}=14.92$; $p<0.001$; $\eta^2=0.21$). BAI was mainly influenced by gender, with females ($p<0.001$) and long-distance swimmers ($p<0.05$) showing higher values, and distance having a stronger effect in females (gender $F_{1,56}=51.72$; $p<0.001$; $\eta^2=0.48$; distance $F_{1,56}=4.25$; $p=0.044$; $\eta^2=0.07$; interaction $F_{1,56}=8.60$; $p=0.005$; $\eta^2=0.13$).

**Table 2. Somatic characteristics of the athletes, mean ± SD.**

| Measurement | SDF (n=24) | LDF (n=6) | SDM (n=24) | LDM (n=6) |
| --- | --- | --- | --- | --- |
| Body height, cm | 173.6±5.3 | 174.0±4.7 | 185.7±5.8 | 184.9±4.4 |
| Body mass, kg | 58.3±5.8[a] | 67.0±4.2 | 78.5±5.7 | 80.8±3.8 |
| Waist circumference, cm | 71.1±3.9[b] | 78.3±5.1 | 82.4±5.8 | 83.2±3.8 |
| Hip circumference, cm | 94.7±4.5[b] | 102.6±3.1 | 97.7±5.1 | 95.7±4.0 |

Abbreviations: SDF - short-distance female swimmers, LDF - long-distance female swimmers, SDM - short-distance male swimmers, LDM - long-distance male swimmers.

[a]significant differences between short-distance and long-distance female swimmers, $p<0.01$.

[b]significant differences between short-distance and long-distance female swimmers, $p<0.001$.

**Table 3. Comparison of anthropometric indexes of female and male swimmers training for short and long distances, mean ±SD.**

| | SDF (n=24) | LDF (n=6) | SDM (n=24) | LDM (n=6) | 'gender' | 'distance' | 'gender' x 'distance' |
|---|---|---|---|---|---|---|---|
| BMI, kg/m$^2$ | 19.3±1.4 | 22.1±0.6 | 22.8±0.8 | 23.1±0.8 | *** | *** | *** |
| BF, % | 22.7±1.6 | 26.2±0.9 | 16.5±1.0 | 16.8±0.9 | *** | *** | *** |
| BAI, cm/m$^{1.5}$ | 23.4±2.0 | 26.7±1.5 | 20.6±2.2 | 20.1±1.8 | *** | * | ** |
| Slenderness index, cm/kg$^{1/3}$ | 44.8±1.1 | 42.9±0.5 | 43.4±0.6 | 43.1±0.6 | * | *** | ** |
| WHR | 0.7±0.0 | 0.8±0.0 | 0.8±0.1 | 0.9±0.0 | *** | NS | NS |

Abbreviations: SDF - short-distance female swimmers, LDF - long-distance female swimmers, SDM - short-distance male swimmers, LDM - long-distance male swimmers; BMI - Body Mass Index, BF - Body Fat, BAI - Body Adiposity Index, WHR - Waist-to-Hip ratio.

* $p<0.05$, ** $p<0.01$, *** $p<0.001$ – statistical significance of the effects; NS – statistically non-significant difference.

In contrast, the Slenderness Index was lower in males and long-distance swimmers (gender $F_{1,56}=5.06$; $p=0.028$; $\eta^2=0.08$; distance $F_{1,56}=17.03$; $p=0.0001$; $\eta^2=0.23$). A significant interaction effect of 'gender x distance' factors ($F_{1,56}=10.89$; $p=0.002$; $\eta^2=0.16$) revealed that SDF had significantly higher Slenderness Index values compared to the other groups ($p<0.001$).

Our results demonstrated that WHR values depended primarily on the gender of the athletes ($F_{1,56}=38.08$; $p<0.001$; $\eta^2=0.40$). Males had significantly higher WHR values ($p<0.001$). No significant effect was found for the factor 'distance' ($F_{1,56}=1.31$; $p=0.26$; $\eta^2=0.02$) and for the interaction 'gender x distance' ($F_{1,56}=0.14$; $p=0.74$; $\eta^2=0.003$) (Table 3).

It was noted that the predominant type of fat distribution in the female and male swimmers studied was the gynoid type. Only 10% of the female athletes and 3% of the male athletes were characterised by android body fat distribution. Gender and distance specialisation did not significantly affect the prevalence of android and gynoid fatness among the athletes studied (G=1.52; $p=0.68$; R2=0.16).

Based on the correlation analysis, we noted a significant positive correlation between FINA points and BMI and WHR values. In contrast, female and male athletes characterised by higher FINA scores obtained lower BF, BAI and Slenderness index values. After taking into account the distance trained, these relationships remained statistically significant primarily in the group of short-distance swimmers (Table 4).

## Discussion

The main aim of this cross-sectional research was to characterise the physique of female and male swimmers compared to their non-training peers and to determine to what extent it is influenced by training distance. The study was conducted among top Polish athletes, whose somatic constitution was analysed on the basis of selected anthropometric indexes and

**Table 4. Spearman correlation coefficient for somatic indexes and FINA points in male and female swimmers specialising in short and long distances.**

| | BMI | BF | BAI | Slenderness index | WHR |
|---|---|---|---|---|---|
| All swimmers (n=60) | | | | | |
| FINA | 0.457* | −0.308* | −0.300* | −0.325* | 0.328* |
| Short-distance swimmers (n=48) | | | | | |
| FINA | 0.429* | −0.438* | −0.319* | −0.241 | 0.353* |
| Long-distance swimmers (n=12) | | | | | |
| FINA | 0.393 | −0.312* | −0.313 | −0.068 | 0.158 |

* $p<0.05$.

comparison with non-training young adults. The general findings of our study are as follows: swimming maintains normal body fat content and health-promoting body fat distribution; the distance trained differentiates especially the physique of female swimmers; and physique has a greater impact on athletic performance in short-distance pool swimmers.

Based on BMI values, we found that 89% of the swimmers were of normal weight (BMI between 18.5 and 24.9). However, in the group of short-distance female swimmers 29% were underweight (BMI < 18.5). The BMI values of the athletes, regardless of the distance trained, were significantly lower than the BMI of the non-athletes. This effect was certainly influenced primarily by body height, which was significantly higher in the short- and long-distance swimmers of both sexes than in the control group. In short-distance female swimmers the significantly lower body mass also contributed to this. In the case of body mass, our results differ from those available in the literature, according to which short-distance swimmers are characterised by higher body mass, mainly due to more developed muscle mass than long-distance swimmers. However, in our participants' greater body mass of long-distance swimmers was associated with higher body fat, as will be discussed below.

The relationship between body height and better swimming performance has been confirmed by many other researchers. Dopsaj et al. [25] also studied relations between swimming performance and the characteristics of body composition in highly trained swimmers. The research included 82 swimmers of international level (N = 46 male and N = 36 female athletes) from 8 European countries. The mean values of the basic anthropometric parameters, i.e., height, body mass and BMI recorded in this group of athletes were similar to our study and were 183.8 cm, 78.4 kg and 23.21 kgm2 for male swimmers and 171.5 cm, 63.1 kg and 21.45 kgm2 for female swimmers, respectively [25]. The aforementioned researchers further highlighted that female swimmers are taller, have lower body weight and BMI comparing with the results published previously [26]. Pla et al. [27] analysing height, weight, BMI and speed results for the top 100 international men and women over distances from 50 to 1,500m freestyle in the 2000–2014 seasons found that height was always positively correlated with speed with 95% probability. In this study, the authors proved that body height has a huge impact on swimming speed at all distances and for both genders. As an explanation, it is given that this is related to a greater arm span and a reduction of the drag coefficient [28].

Swimmers, compared to non-swimmers, were also characterised by significantly lower WHR values, which was a consequence of a significantly smaller waist circumference and a significantly larger hip circumference. The WHR values of both female and male swimmers indicated a gynoid type of fatness of their bodies. In explaining the benefits of this type of fat distribution in swimmers, it seems right to refer to the specific environment in which the exercise is performed. During swimming, the water around the swimmer is set in motion, which can be thought of as additional water mass to be overcome. People with a lower WHR have a lower added mass, which significantly reduces the energy cost of exercise. Similarly explained are the differences in the energy cost of swimming between males and females, who also differ in WHR values [29]. The gynoid type of fatness that characterises swimmers should also be considered beneficial from a health point of view. This type of fatness is characterised by less visceral fat compared to subcutaneous fat, which significantly reduces the risk of non-communicable diseases such as hypertension, diabetes, coronary heart disease and dyslipidaemia [30].

We observed that swimmers, irrespective of the distance trained, had significantly lower levels of body fat as expressed by the BF index compared to their untrained peers. Relating the swimmers' BF values to the healthy body fat norms proposed by Gallagher et al. [23], it should be noted that the values were within the ranges for lean individuals, for both women (21–33%) and men (8–21%). However, when comparing the fat percentage of the athletes participating in own study with the norms for swimmers proposed by Santos et al. [26], we found that the short-distance female swimmers had a BF at the 50th percentile (% body fat for this percentile in females is 23.29), while the long-distance ones were above the 75th percentile (% body fat for this percentile in females is 25.15). Among male swimmers, regardless of the distance trained, the average body fat content exceeded the 75th percentile value (% body fat for this percentile in males is 15.02) for representatives of this sport determined by Santos et al. [26]. BAI values, the second fatness index used, did not differ

between swimmers and their non-training peers. BAI indicated a higher body fat content than that indicated by BF, which is related to the fact that the calculation of BAI takes into account hip circumference, whose values in the swimmers, regardless of the distance trained, were higher than in the control group.

We also found that swimmers had significantly higher Slenderness index values compared to those who did not train. Relating the values of this index to norms, we concluded that among the short-distance female swimmers, 54% had a medium build (index values between 41.48–44.96) and 46% had a slender build (index values > 44.97), while all the long-distance female swimmers were characterised by a medium build. The Slenderness index values of the male swimmers, regardless of the distance trained, indicated a medium build.

This study also aimed to determine whether distance specialisation significantly differentiates the body physique of top male and female swimmers. Our results showed that such a relationship occurs mainly in female swimmers. Similar results were obtained by other researchers analysing the relationship between performance, body composition and somatotype in competitive swimmers [25]. Based on these findings, the authors stated that body composition and somatotype characteristics could serve as predictors of performance only for female athletes.

In fact, it turned out that short-distance female swimmers were characterised by significantly lower body weight, waist and hip circumferences, as well as lower body fat content compared to long-distance ones. So they had a slimmer physique. This is in agreement with reports by other authors on the relationship between body build and swimming speed. Namely, there are significant evidence that fat reduction contributes to the development of speed and agility [31]. Lower body fat content probably results in lower body shape resistance (frontal area) and skin friction resistance. On the other hand, because body fat has a lower density than water, a higher body fatness may be beneficial due to increased buoyancy in water. This could explain the significantly higher body fat content in long-distance female swimmers than in short-distance ones.

Another reason for the higher body fat content in long-distance swimmers is the higher energy cost of prolonged exercise. The most energy-efficient substrate is fat, which, stored in subcutaneous adipose tissue, can additionally act as an insulating layer. A number of studies have shown that swimmers with more subcutaneous fat were able to withstand longer periods of cold water swimming compared to those with less body fat [32]. Among others, Scott et al. [33] observed that the range of mean subcutaneous adipose tissue thickness of competitive swimmers was 6–10 mm compared to an average value of 4 mm reported in control subjects. However, the relationship between body fat and improved swimming performance appears more complex. Roelofs et al. [34] observed that changes in % body fat during the competitive season of sprinters, distance swimmers and divers were not associated with changes in their performance. They showed that better performance of distance swimmers was significantly correlated with lean mass (those who had a greater increase in lean mass improved their performance more). This demonstrates the need for further research among swimmers training at different distances.

Correlation analysis between the body composition indexes and FINA scores confirmed that the relationship between body composition and athletic performance in swimmers depends on the distance trained. The statistically significant correlations recorded in the whole group after its division by trained distance were essentially maintained in the group of short-distance swimmers. This seems to mean that in long-distance swimming, success is determined more by factors other than the athlete's physique. At the same time, it should be emphasised that the lack of this correlation in the group of long-distance swimmers is due to their small numbers. However, during the course of the presented study, so many long-distance competitors met the criteria for inclusion in the study, i.e., participation in the national championships and consent to participate in the study.

Our study has several limitations. The first of these is the aforementioned smaller number of long-distance swimmers compared to short-distance ones. Another limitation is the use of calculated body fat to assess swimmers' physiques. However, this is how the amount of body fat was assessed in the population that constituted the control group in the present study. In future studies on the relationship between distance trained and swimmers' physique, it would furthermore be worth considering style specialisation. Swimming style can affect body composition, as different styles require different

amounts of energy and involve different training patterns. For example, freestyle and butterfly strokes are generally less energy-intensive than the breaststroke, which may result in higher body fat content in athletes specialising in these strokes. Therefore, further research is needed, in which, in addition to the distance covered, the training style will also be taken into account in the analysis of body composition.

Summarising the results of the study, it must be said that explaining the relationship between swimmers' body structure and the distance trained is complicated. This is because it requires taking into account not only the duration and intensity of the training, but also many factors related to the environment, including thermal conditions, drag coefficient, underwater currents, passive resistance, etc. factors that determine the energy cost of the effort. However, conducting such studies, especially if the participants are top athletes, provides important information about the impact of training on their physique, and thus on their sporting performance. This kind of analysis also document the health benefits of swimming, allowing this form of physical activity to be promoted to the public.

## Author contributions

**Conceptualization:** Aleksandra Stachowicz, Anna Kęska, Małgorzata Stachowicz.

**Data curation:** Aleksandra Stachowicz, Anna Kęska.

**Formal analysis:** Katarzyna Milde.

**Funding acquisition:** Aleksandra Stachowicz, Anna Kęska.

**Investigation:** Aleksandra Stachowicz, Małgorzata Stachowicz.

**Methodology:** Aleksandra Stachowicz, Małgorzata Stachowicz.

**Project administration:** Aleksandra Stachowicz, Anna Kęska, Małgorzata Stachowicz.

**Resources:** Aleksandra Stachowicz, Małgorzata Stachowicz.

**Software:** Aleksandra Stachowicz, Katarzyna Milde.

**Supervision:** Aleksandra Stachowicz, Anna Kęska.

**Validation:** Aleksandra Stachowicz, Anna Kęska.

**Visualization:** Aleksandra Stachowicz, Anna Kęska, Katarzyna Milde, Małgorzata Stachowicz.

**Writing – original draft:** Aleksandra Stachowicz, Anna Keska.

**Writing – review & editing:** Aleksandra Stachowicz, Anna Kęska.

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
