## [Decision Letter · Decision Letter 0]

8 Aug 2025

PONE-D-25-30969Comparing the physiques of elite Polish female and male swimmers training for short and long distances with their non-training peers - is swimming a health-promoting sport?PLOS ONE?

Dear Dr. Keska,

Thank you for submitting your manuscript to PLOS ONE. After careful consideration, we feel that it has merit but does not fully meet PLOS ONE’s publication criteria as it currently stands. Therefore, we invite you to submit a revised version of the manuscript that addresses the points raised during the review process.

We look forward to receiving your revised manuscript.

Kind regards,

Emiliano Cè, Ph.D.

Academic Editor

PLOS ONE

Journal Requirements: 

Reviewers' comments:

Reviewer's Responses to Questions

**Comments to the Author**

1. Is the manuscript technically sound, and do the data support the conclusions?

Reviewer #1: Yes

Reviewer #2: Partly

2. Has the statistical analysis been performed appropriately and rigorously?

Reviewer #1: Yes

Reviewer #2: Yes

3. Have the authors made all data underlying the findings in their manuscript fully available?

Reviewer #1: Yes

Reviewer #2: Yes

4. Is the manuscript presented in an intelligible fashion and written in standard English?

Reviewer #1: Yes

Reviewer #2: Yes

Reviewer #1: The manuscript presents a relevant and timely investigation into the anthropometric characteristics of elite Polish swimmers, comparing them with non-training peers and exploring the influence of training distance and gender. The topic is both scientifically interesting and practically valuable, especially in the context of understanding the health-promoting aspects of competitive swimming.

However, several areas need revision before the manuscript is suitable for publication:

Sample size imbalance:

The long-distance swimmer subgroups (n=6 each) are underpowered, limiting the generalizability of some conclusions. The small sample size should be acknowledged more clearly in both the discussion and limitations section. Consider discussing whether any power analysis was conducted.

Body fat estimation:

The use of formula-based estimations (e.g., BF%, BAI) instead of direct measures like DEXA or BIA may reduce accuracy. While this limitation is mentioned, it should be more clearly emphasized, especially regarding how it might affect comparisons with reference norms.

Swimming stroke not accounted for:

Different strokes (freestyle, breaststroke, etc.) may influence body composition differently. Although this is briefly mentioned in the limitations, adding a sentence on how stroke style could confound findings would strengthen the manuscript.

Clarity of subgroup comparisons:

In several parts of the results and discussion, findings from male and female swimmers, or short- and long-distance groups, are compared. It would help the reader if the authors clarified whether these comparisons are statistically supported or descriptive.

Figure and table presentation:

Ensure that figure legends clearly indicate significance levels and group comparisons. For example, Figure 1 uses symbols (^, #, *) that could be better explained for clarity.

Reviewer #2: Dear Authors,

I want to express my gratitude for the opportunity to review this manuscript.

It's an interesting and important topic in the field of sport, particularly in swimming.

At this stage, the document requires improvements, below with line indication:

31-34 – Please consider presenting the “p” in italics and presenting more numerical results in the abstract.

41-82 - Please consider developing the introduction section, supporting the text with references, and clearly highlighting the research gap before the presentation of the aims of the study. Please consider standardizing the paragraph size to improve readability (8-12 lines suggested throughout the manuscript).

91 – Please revise the citation format (not only in this line, throughout the manuscript).

84-148 - Please consider characterizing the sample. Some examples: Swimmer's training routines (in-water / dryland) and background. It is also relevant to describe the inclusion and exclusion criteria. Moreover, all data collection and statistical procedures should also be described in detail – e.g. sample power (GPower used?).

141-241 – Please consider reformulating the results section. It is too long, which difficult for readers' comprehension.

243-262 - Please consider shorter paragraphs in this section to improve readability, addressing the study rationale, and at the end of the section, presenting the study limitations and suggestions for future research.

381 – Please double-check the references format.

Please consider improving the English details throughout the manuscript.

Please consider improving the quality of the figures.

**Do you want your identity to be public for this peer review?** For information about this choice, including consent withdrawal, please see our Privacy Policy

Reviewer #1: **Yes: ** Dr. Mansour Sahebozamani

Reviewer #2: **Yes: ** Mário Espada

---

## [Author Response · Author response to Decision Letter 1]

19 Sep 2025

Response to Reviewer 1

Dear Reviewer,

We sincerely appreciate your comprehensive review and constructive feedback on our manuscript. Your comments have been extremely helpful in identifying areas where our manuscript needs improvement.

Below are the specific responses and explanations of the revisions for each of the comments raised.

1. Sample size imbalance: The long-distance swimmer subgroups (n=6 each) are underpowered, limiting the generalizability of some conclusions. The small sample size should be acknowledged more clearly in both the discussion and limitations section. Consider discussing whether any power analysis was conducted.

We fully agree with the reviewer's comment that the small number of long-distance swimmers is a limitation of this study. This comment has been added to the section ‘limitations of this study’ in the manuscript. At the same time, we would like to clarify that among top Polish swimmers, who were the only ones included in the study, there were significantly fewer people training for long distances. This distance is less popular not only in Poland. We would also like to emphasise that the small number of long-distance swimmers was due to compliance with the criteria for inclusion in the study. The participants were swimmers who competed in the national championships during the study and agreed to take part in the research.

2. Body fat estimation: The use of formula-based estimations (e.g., BF%, BAI) instead of direct measures like DEXA or BIA may reduce accuracy. While this limitation is mentioned, it should be more clearly emphasized, especially regarding how it might affect comparisons with reference norms.

Once again, we fully agree with the reviewer's opinion that calculating body fat content is a much less accurate method than measuring it using methods such as BIA or DEXA. With this in mind, we used the standards proposed by Gallagher et al. to interpret the results obtained. These authors established percentage body fat levels corresponding to BMI thresholds for underweight (<18.5), overweight (≥25) and obesity (≥30) based on gender-adjusted formulas that estimate relative body fat content based on BMI and other potential independent variables such as age and ethnicity. These formulas have been validated using reference methods for assessing total body fat content.

3. Swimming stroke not accounted for: Different strokes (freestyle, breaststroke, etc.) may influence body composition differently. Although this is briefly mentioned in the limitations, adding a sentence on how stroke style could confound findings would strengthen the manuscript.

Thank you for pointing this out. We have added this sentence to the limitations of the study section.

Swimming style can affect body composition, as different styles require different amounts of energy and involve different training patterns. For example, freestyle and butterfly are typically less energy-intensive than breaststroke, which may result in higher body fat content in athletes who specialise in these styles. Therefore, further research is needed, in which, in addition to the distance covered, the training style will also be taken into account in the analysis of body composition.

4. Clarity of subgroup comparisons: In several parts of the results and discussion, findings from male and female swimmers, or short- and long-distance groups, are compared. It would help the reader if the authors clarified whether these comparisons are statistically supported or descriptive.

Any differences between the compared subgroups, distinguished both in terms of the gender of the study participants and the trained distance, were statistically verified.

Once again, we thank you for your valuable time and feedback. We look forward to your further comments and hope that our revisions meet your expectations.

Response to Reviewer 2

We are truly grateful for your thorough review and valuable suggestions regarding our manuscript. Your feedback has been instrumental in highlighting aspects that required refinement.

Below, we provide detailed responses and explanations of the revisions made in accordance with each of your comments.

1. Please consider presenting the “p” in italics and presenting more numerical results in the abstract.

Thank you for this tip. We have made the appropriate changes to the manuscript.

2. Please consider developing the introduction section, supporting the text with references, and clearly highlighting the research gap before the presentation of the aims of the study. Please consider standardizing the paragraph size to improve readability (8-12 lines suggested throughout the manuscript).

As suggested by the reviewer, we have added information to the Introduction justifying the purpose of the research and we have revised the length of the paragraphs throughout the manuscript.

3. Please revise the citation format (not only in this line, throughout the manuscript).

Thank you for this tip. We have made the appropriate changes to the manuscript.

4. Please consider characterizing the sample. Some examples: Swimmer's training routines (in-water / dryland) and background. It is also relevant to describe the inclusion and exclusion criteria. Moreover, all data collection and statistical procedures should also be described in detail – e.g. sample power (GPower used?).

Following the reviewer's advice, we have supplemented the information on the participants' training and the criteria for inclusion and exclusion from the study.

Over the study period, all athletes were completing 8-10 in-water training sessions a week (90–120 minutes), swimming a distance of 6,000–14,000 m per day. They also underwent land training 2 to 4 times a week (60–90 minutes), during which they performed strength, mobility and stabilisation exercises.

The inclusion criteria were high athletic level and participation in training sessions during the study, while exclusion criteria were injury preventing training or lack of consent to participate in the study.

The participants underwent identical assessment session in the morning hours. Before the measurements, they were well rested, after fasting overnight, and properly hydrated. They were asked to avoid intense physical activity for at least 12 hours before the measurements. All tests were conducted in a well-ventilated test room with controlled temperature and humidity by a trained person with many years of experience.

5. Please consider reformulating the results section. It is too long, which difficult for readers' comprehension.

Thank you for bringing this to our attention. The description of the results has been shortened. We hope that in its current form it is more comprehensible.

6. Please consider shorter paragraphs in this section (Discussion) to improve readability, addressing the study rationale, and at the end of the section, presenting the study limitations and suggestions for future research.

The Discussion section has been modified in accordance with the reviewer's suggestions.

7. Please double-check the references format.

The formatting of the references has been corrected.

Once again, we sincerely thank you for your thoughtful and constructive feedback. We hope that the revisions made will meet the reviewer’s expectations and lead to the acceptance of our manuscript.

On behalf of all authors,

yours sincerely,

Anna Kęska

---

## [Decision Letter · Decision Letter 1]

26 Oct 2025

PONE-D-25-30969R1Comparing the physiques of elite Polish female and male swimmers training for short and long distances with their non-training peers - is swimming a health-promoting sport?PLOS ONE?

Dear Dr. Keska,

Thank you for submitting your manuscript to PLOS ONE. After careful consideration, we feel that it has merit but does not fully meet PLOS ONE’s publication criteria as it currently stands. Therefore, we invite you to submit a revised version of the manuscript that addresses the points raised during the review process.

**ACADEMIC EDITOR:****Dear Editor, **plosone@plos.org . A rebuttal letter that responds to each point raised by the academic editor and reviewer(s). You should upload this letter as a separate file labeled 'Response to Reviewers'.A marked-up copy of your manuscript that highlights changes made to the original version. You should upload this as a separate file labeled 'Revised Manuscript with Track Changes'.An unmarked version of your revised paper without tracked changes. You should upload this as a separate file labeled 'Manuscript'.

We look forward to receiving your revised manuscript.

Kind regards,

Emiliano Cè, Ph.D.

Academic Editor

PLOS ONE

Journal Requirements:

Reviewers' comments:

Reviewer's Responses to Questions

**Comments to the Author**

Reviewer #2: All comments have been addressed

2. Is the manuscript technically sound, and do the data support the conclusions?

Reviewer #2: Partly

3. Has the statistical analysis been performed appropriately and rigorously?

Reviewer #2: Yes

4. Have the authors made all data underlying the findings in their manuscript fully available?

Reviewer #2: Yes

5. Is the manuscript presented in an intelligible fashion and written in standard English?

Reviewer #2: Yes

Reviewer #2: Dear Authors,

Thank you for considering my suggestions and incorporating them into the manuscript, which has been globally improved. Congratulations.

Below are some specific suggestions with line indications.

L234 – Please place the units in the variables and describe the variables in full in the table footnote.

L283-284 – Please revise the “p” format – Suggested in lowercase and italics throughout the manuscript. Please carefully check all format details throughout the manuscript, considering the journal template and instructions for authors..

L317 – Please consider placing the citation number close to the name of the authors. Please revise all manuscript considering this detail.

L331-335 – Please consider standardizing the paragraphs size (8-12 lines suggested), aiming to improve readability. In these lines the paragraphs is too short, in other too long.

L501 – Please revise ref 17 format. Please carefully revise all references format.

**Do you want your identity to be public for this peer review?** For information about this choice, including consent withdrawal, please see our Privacy Policy

Reviewer #2: No

---

## [Author Response · Author response to Decision Letter 2]

22 Nov 2025

Response to Reviewer 2

Dear Reviewer,

Thank you very much for your further comments on our manuscript. Below are the specific responses and explanations of the revisions for each of the comments raised.

1. L234 – Please place the units in the variables and describe the variables in full in the table footnote.

Thank you for this tip. We have made the appropriate changes to the manuscript.

2. L283-284 – Please revise the “p” format – Suggested in lowercase and italics throughout the manuscript. Please carefully check all format details throughout the manuscript, considering the journal template and instructions for authors.

As suggested by the reviewer, we have corrected format of ‘p’ throughout the manuscript.

3. L317 – Please consider placing the citation number close to the name of the authors. Please revise all manuscript considering this detail.

Thank you for this suggestion. We have made the appropriate changes to the manuscript.

4. L331-335 – Please consider standardizing the paragraphs size (8-12 lines suggested), aiming to improve readability. In these lines the paragraphs is too short, in other too long.

Following the reviewer's advice, we have attempted to standardise the length of the paragraphs while maintaining their substantive consistency.

5. L501 – Please revise ref 17 format. Please carefully revise all references format.

Thank you for this tip. We have rechecked and corrected all references.

Thank you very much for these comments. We hope that the revised manuscript will be accepted by the reviewer.

On behalf of all authors,

yours sincerely,

Anna Kęska

---

## [Decision Letter · Decision Letter 2]

1 Dec 2025

Comparing the physiques of elite Polish female and male swimmers training for short and long distances with their non-training peers - is swimming a health-promoting sport?

PONE-D-25-30969R2

Dear Dr. Keska,

We’re pleased to inform you that your manuscript has been judged scientifically suitable for publication and will be formally accepted for publication once it meets all outstanding technical requirements.

Kind regards,

Emiliano Cè, Ph.D.

Academic Editor

PLOS ONE

Additional Editor Comments (optional):

Reviewers' comments:

Reviewer's Responses to Questions

**Comments to the Author**

Reviewer #2: All comments have been addressed

2. Is the manuscript technically sound, and do the data support the conclusions?

Reviewer #2: Yes

3. Has the statistical analysis been performed appropriately and rigorously?

Reviewer #2: Yes

4. Have the authors made all data underlying the findings in their manuscript fully available?

Reviewer #2: Yes

5. Is the manuscript presented in an intelligible fashion and written in standard English?

Reviewer #2: Yes

Reviewer #2: Dear Authors,

Thank you and congratulations on your work during the review process, addressing all of the reviewers' comments.

Best wishes for the continued development of your good work.

Best regards,

**Do you want your identity to be public for this peer review?** For information about this choice, including consent withdrawal, please see our Privacy Policy

Reviewer #2: **Yes: ** Mário André da Cunha Espada

---

## [Editor Report · Acceptance letter]

PONE-D-25-30969R2

PLOS One

Dear Dr. Keska,

I'm pleased to inform you that your manuscript has been deemed suitable for publication in PLOS One. Congratulations! Your manuscript is now being handed over to our production team.

Kind regards,

on behalf of

Prof. Emiliano Cè

Academic Editor

PLOS One